# The biopsychosocial factors associated with development of chronic musculoskeletal pain. An umbrella review and meta-analysis of observational systematic reviews

**Michael Dunn** [1,2,3]*, **Alison B. Rushton**[1,4], **Jai Mistry**[2,4], **Andrew Soundy**[1], **Nicola R. Heneghan**[1]

**1** Centre of Precision Rehabilitation for Spinal Pain, School of Sport and Exercise Science, University of Birmingham, Birmingham, West Midlands, United Kingdom, **2** Musculoskeletal Physiotherapy, St. George's University Hospitals NHS Foundation Trust, London, United Kingdom, **3** Institute of Medical and Biomedical Education, Centre for Allied Health, St. George's University of London, London, United Kingdom, **4** School of Physical Therapy, Western University, London, Ontario, Canada

* mdunn@sgul.ac.uk

**Data Availability Statement:** All relevant data are within the manuscript and its Supporting Information files.

## Abstract

### Aim

The aim of this umbrella review was to establish which biopsychosocial factors are associated with development of chronic musculoskeletal pain.

### Methods

Ovid Medline, Embase, Web of Science Core Collection, Cochrane Database of Systematic Reviews, Database of Abstracts of Reviews of Effects, PsycINFO, CINAHL, PEDro, PROSPERO, Google Scholar and grey literature were searched from database inception to 4th April 2023. Systematic reviews of observational prospective longitudinal studies, including populations with <3 months (not chronic) musculoskeletal pain, investigating biopsychosocial factors that contribute to development of chronic (>3 months) musculoskeletal pain. Two reviewers searched the literature, assessed risk of bias (Assessing the Methodological Quality of Systematic Reviews-2), and evaluated quality (Grading of Recommendations, Assessment, Development and Evaluation) to provide an overall statement on the certainty of evidence for each biopsychosocial factor. Data analysis was performed through random effects meta-analysis (including meta-analysis of meta-analyses where possible) and descriptive synthesis.

### Results

13 systematic reviews were included comprising 185 original research studies (n = 489,644 participants). Thirty-four biopsychosocial factors are associated with development of chronic musculoskeletal pain. Meta-analyses of odds and/or likelihood ratios were possible for 25 biopsychosocial factors. There is moderate certainty evidence that smoking (OR 1.24 [95%CI, 1.14–1.34), fear avoidance (LR+ 2.11 [95%CI, 1.59–2.8]; LR- 0.5 [95%CI, 0.35–

**Funding:** The author(s) received no specific funding for this work.

**Competing interests:** The authors have declared that no competing interests exist.

0.71]) poorer support networks (OR 1.21 [95%CI, 1.14–1.29]), lower socioeconomic status (OR 2.0 [95%CI, 1.64–2.42]), and high levels of pain (OR 5.61 [95%CI, 3.74–8.43]) are associated with development of chronic musculoskeletal pain (all P<0.001). Remaining factors are of low or very low certainty evidence.

## Conclusions and relevance

There is moderate certainty evidence that smoking, fear avoidance, poorer support networks, lower socioeconomic status, and high levels of pain are associated with development of chronic musculoskeletal pain. High risk of bias was evident in most included reviews; this highlights the need for higher quality systematic reviews.

## Introduction

The International Classification of Diseases describes chronic musculoskeletal pain (CMP) as pain arising from bones, joints, muscle or related soft tissues lasting longer than three months [1]. The burden of CMP to individuals and societies is substantial being the greatest cause of disability worldwide affecting approximately 22% of the global population [2]. Once CMP is established it is hard to treat with 79–92% of people still experiencing CMP up to 12 years later [3–5]. Consequently, CMP is the most common cause of sickness absence from work (after common minor illnesses) [6] and only 59% of the working age population are in work [7]. The personal burden of CMP is also substantial with many individuals experiencing moderate to severe disability [8], poorer quality of life [9], and higher risk of chronic diseases including cardiovascular disease, diabetes and cancer [10]. Despite the United Kingdom (UK) National Health Service spending £5 billion every year on treating musculoskeletal (MSK) pain [6], the prevalence of CMP is rising [11]. These points illustrate the huge burden on individuals and society and suggest that current healthcare management of CMP may benefit from a refined approach.

Acute episodes of MSK pain are a common experience across individuals where pain and dysfunction typically subsides within three months coinciding with healing of injured or irritated MSK structures [12]. The mechanisms of CMP are different to acute pain in that pain exists despite there no longer being evidence of ongoing healing, but rather due to a sensitised nervous system that creates a continued or repeated experience of pain despite no evidence of actual or potential tissue damage [13, 14]. This transition from acute to chronic MSK pain is associated with the presence of many biopsychosocial factors such as fear avoidance, low mood, and work satisfaction or strain [15–17]. Despite this, healthcare services conventionally utilise approaches to treat CMP based on understandings of acute MSK pain, with focus often on identifying and treating perceived injured or irritated MSK structures. This does not take into account the complexity of CMP; rather, these approaches are grounded in simple mechanistic theories (e.g., debridement of degenerative joints) and traditional observational evidence [18]. However, contemporary higher quality research, such as randomised placebo-controlled trials, demonstrates that many approaches based on treating MSK structures in CMP are no better than placebo with many common orthopaedic surgeries now known to be only equally as efficacious as sham surgery [19, 20]. Furthermore, many of the changes observed through radiographic imaging previously thought to explain CMP are now known to be highly prevalent in people with no history of pain [21, 22]. These points demonstrate that purely structural based approaches to managing CMP are simplistic.

Despite these advancing understandings, many clinicians still employ MSK structural based approaches to treating CMP [23] with biopsychosocial approaches typically only endorsed after these have been unsuccessful [24]. But CMP is difficult to treat once it is established and therefore biopsychosocial approaches used at this late stage may be of limited benefit. However, if utilised during acute MSK pain, it is possible that biopsychosocial approaches could prevent development of CMP. This theory is informed by many prospective longitudinal studies summarised by systematic reviews which identifies a number of biopsychosocial factors that are present during acute MSK pain and associated with transition to CMP [16, 25, 26]. Early identification of these factors would provide the opportunity for proactive, preventative healthcare approaches; a strategy that works well for other chronic diseases such as heart disease [27] and diabetes [28].

To inform proactive biopsychosocial management aimed at preventing CMP, a clear understanding of the biopsychosocial factors that contribute to its development is needed. There are a number of systematic reviews which have investigated this for specific MSK conditions (e.g., back pain), however many of the biopsychosocial factors identified are not related to a condition but rather are characteristics of the person and/or their experience of pain (e.g., fear avoidance, severe pain etc). It is therefore possible that these biopsychosocial factors transcend specific forms of MSK pain (e.g., back pain) and are relevant for all types of MSK pain, but this is not clear from existing evidence. It is therefore timely to perform an umbrella review to aggregate findings of systematic reviews of biopsychosocial factors associated with development of CMP that are relevant for all MSK conditions.

## Methods

### Aim

The aim of this umbrella review was to identify which biopsychosocial factors are associated with development of CMP.

### Design

An umbrella review informed by the Joanna Briggs Institute Manual for Evidence Synthesis of Umbrella Reviews [29] and the Cochrane handbook for the conduct of systematic reviews [30], registered in PROSPERO (CRD42020193081) and protocol published *a priori* [31], is reported in adherence to the Preferred Reporting Items for Systematic Reviews and Meta-Analyses (PRISMA) checklist [32] (see S1 File).

### Eligibility criteria

  **Inclusion criteria.**

- **Population:** adults (>18) with <3 months of MSK pain.

- **Exposure:** individuals' experience of any biopsychosocial factors (e.g., smoking).

- **Comparator:** individuals who do not experience the biopsychosocial factor under investigation (e.g., non-smoker).

- **Outcome:** MSK pain >3 months identified through any patient reported outcome measure.

- **Study designs:** systematic reviews, with or without meta-analysis, of observational prospective longitudinal studies (the gold standard for epidemiological research [33]). Original studies included in reviews must have been at least three months in duration with no limitations on the study setting.

**Exclusion criteria.** Systematic reviews which include interventional studies (e.g., factors associated with successful surgery), populations with other plausible explanations for CMP (e.g., autoimmune disorders), injuries where tissue healing may be incomplete at three months (e.g., fractures), draw body region specific conclusions which are not generalisable to the wider CMP population (e.g., a bony heel spur), pool data with non-MSK chronic pain populations (e.g., cancer related pain), and systematic reviews where the full text was not available in the English language. No limitations were placed on eligibility based on review quality or that of included original studies.

## Information sources

We searched MEDLINE, EMBASE, Web of Science Core Collection, PEDro, CINAHL, PsycINFO, Cochrane database for systematic reviews, the Database of Abstracts of Reviews of Effects, Google Scholar, the PROSPERO register. There was no limitation on search dates with final searches performed on 4<sup>th</sup> April 2023. Grey literature was searched using the Canadian Agency for Drugs and Technologies in Health grey literature searching tool. To ensure literature saturation, reference lists of included studies or relevant reviews identified through the search were also screened for potentially eligible systematic reviews. See S1 Table for our search strategy designed with Ovid MEDLINE.

## Screening and selection

Two reviewers (MD & JM) independently performed searches and screened titles and abstracts for consideration of full text review using EndNote X9.3.3. Full texts were then sourced and discussed for eligibility of inclusion, with any disagreements referred to a third reviewer (NH). Reviewers were not blinded to the journal titles, study authors or institutions. Reasons for excluding reviews were recorded.

## Data extraction

Two reviewers (MD & JM) independently extracted data. Data was extracted using a standardised proforma which was piloted *a priori* [31] and included review and population characteristics, sample size, biopsychosocial factor details, length of follow-up, and any relevant quantitative or descriptive findings. Data were not extracted from primary research studies but from included reviews only [29]. Review authors were contacted where data were unclear or missing.

## Risk of bias

Two reviewers (MD & JM) independently performed risk of bias assessment using the Assessing the Methodological Quality of Systematic Reviews (AMSTAR) 2 checklist [34] and based on this rated reviews as low, moderate or high risk of bias, with any disagreements referred to a third reviewer (NH). Where included review authors performed risk of bias assessments of primary studies, these were synthesised to provide an overall rating of the risk of bias of primary studies that support each biopsychosocial factor (see Table 1).

## Statistical analysis and data synthesis

Meta-analysis was performed using SPSS 29.0.0.0 where at least two synthesisable sets of quantitative results were reported for the same factor. Effect sizes and 95% confidence intervals were extracted from included reviews. In line with Cochrane Handbook guidance [30], ratio effect sizes were converted to the natural log scale with standard errors computed from 95%

**Table 1. Risk of bias rating of primary studies for each biopsychosocial factor.**

| Risk of Bias of Primary Studies | |
|---|---|
| Low | >75% of studies rated as low risk of bias, or consistent findings across studies and at least 2 low risk of bias studies |
| Moderate | Fails to fulfil low risk of bias criteria, and >50% of studies rated as low or moderate risk of bias |
| High | Fails to fulfil low risk of bias criteria, and >50% of studies rated as high risk of bias, or no risk of bias assessment performed by the included review |

Adapted from Burgess et al. (2020) [35]

confidence intervals. A DerSimonian and Laird inverse variance random effects method was used to compute pooled effect sizes, 95% confidence intervals, Z-value and P-value of statistical significance. Forest plots of findings are presented on the natural log scale (where the number of null effect is 0 rather than 1) to ensure symmetrical representation of 95% confidence intervals [30]. Effect sizes presented in text, tables, and other figures are not presented on the natural log scale to ensure ease of understanding and interpretation for all stakeholders. Where possible, meta-analysis of meta-analyses was performed. Some reviews did not perform meta-analyses but did present synthesisable quantitative findings from primary studies. In this case, a meta-analysis of these findings was performed. Where reviews present both unadjusted and adjusted effect sizes (e.g., for publication bias), the adjusted effect size was used. If there was only one meta-analysis finding and therefore further meta-analysis was not possible, we presented the meta-analysis performed by the primary review in text, figures and tables. A descriptive synthesis was also performed for all findings incorporating findings from all reviews including those not included in meta-analysis.

## Grading of Recommendations Assessment, Development and Evaluation (GRADE)

GRADE is a well-established tool commonly used within systematic reviews to determine certainty of findings, and has been recommended for use with umbrella reviews [36]. To the best of the authors knowledge there is no published guidance for application of GRADE in umbrella reviews, therefore, existing guidance for assessing the five domains of GRADE in reporting the certainty of evidence of prognostic factors were adapted for the purpose of umbrella reviews in collaboration with the lead author of existing GRADE guidance [37]. See S2 File for an overview of GRADE methods used. It was not possible to perform GRADE assessment in instances where factors were supported by only one included review.

Assessment of publication bias was planned with Egger's regression test and visual inspection of a funnel plot, however this was not appropriate due to the number of effect sizes included in each meta-analysis falling below the recommended 10 required for this method [30]. Publication bias was therefore assessed for each included review and biopsychosocial factor in line with Cochrane guidance for umbrella reviews [38] (see S2 File).

## Results

A total of 10,374 studies were screened, with 209 full text articles evaluated for eligibility and 13 systematic reviews included [39–51] summarising 185 primary studies. The number of studies retrieved from each database and the number excluded at each phase of screening and reasons for exclusion are shown in Fig 1. The study characteristics of included reviews are provided in S2 Table.

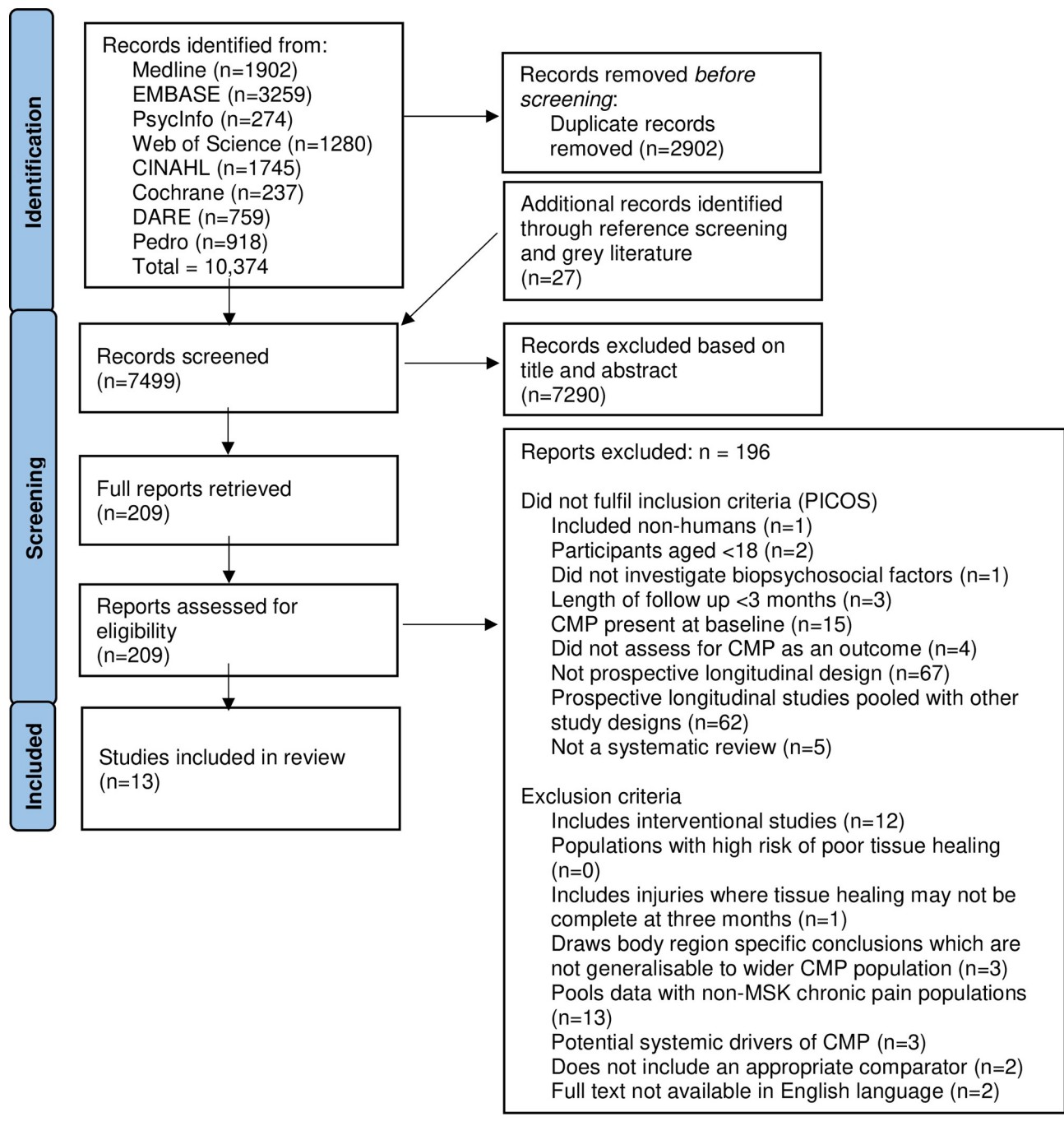

**Fig 1. PRISMA flow diagram.**

### The biopsychosocial factors associated with development of CMP

A total of 92 biopsychosocial items were identified and grouped into 35 biopsychosocial factors. Thirty-four biopsychosocial factors identified are associated with development of CMP, with high body mass index the only factor not associated with development of CMP. The findings have been situated within five overarching domains which can be screened and targeted for intervention in clinical practice: physical health (3 factors), psychological (10 factors),

psychosocial (10 factors), symptoms or experiences at or near onset (9 factors), and demographics (2 factors). Domains and factors were identified by co-authors who are practicing primary care clinicians in the UK NHS (MD & JM). To ensure ease of applicability of findings, a definition for each biopsychosocial domain and factor has been provided based on a synthesis of definitions/descriptions or outcome measures used by included reviews, available in S3 Table.

## Data synthesis

Meta-analysis was possible for 25 factors with odds ratios and/or likelihood ratios. Full details are available in Table 2 and a forest plot representation is available in Fig 2. Of these, 13 odds ratios and 13 likelihood ratios demonstrate statistical significance (P<0.05). A summary of findings including GRADE assessment is available in Table 3. A descriptive synthesis of

**Table 2. Meta-analyses findings.**

| Meta-analysis of meta-analyses (OR) | Studies | Sample | OR (95% CI) | | | Z | P |
|---|---|---|---|---|---|---|---|
| Lower job satisfaction | 39 | 57,794 | 1.43 (1.25–1.63) | | | 5.37 | <0.001 |
| Higher job demands | 77 | 115,148 | 1.25 (1.15–1.35) | | | 5.45 | <0.001 |
| Lower job control | 54 | 82,892 | 1.28 (1.20–1.37) | | | 7.53 | <0.001 |
| Poorer support networks | 69 | 94,954 | 1.21 (1.14–1.29) | | | 6.33 | <0.001 |
| Lower socioeconomic status | 8 | 11,293 | 2.00 (1.64–2.42) | | | 6.95 | <0.001 |
| Female sex/gender | 20 | 6762 | 1.43 (1.13–1.81) | | | 3.01 | 0.003 |
| History of the same MSK pain | 18 | 4803 | 1.24 (0.73–2.12) | | | 0.80 | 0.426 |
| **Meta-analysis (OR)** | | | | | | | |
| Post-trauma stress symptoms | 7 | 1695 | 1.92 (1.37–2.69) | | | 3.776 | <0.001 |
| Catastrophising | 3 | 277 | 3.99 (1.33–10.74) | | | 2.49 | 0.01 |
| Poorer recovery expectations | 6 | 2514 | 2.72 (1.68–4.35) | | | 4.09 | <0.001 |
| Lower job security | 8 | 11,817 | 1.43 (1.16–1.76) | | | 2.33 | <0.01 |
| High levels of pain at or near onset | 11 | 2856 | 5.61 (3.74–8.43) | | | 8.31 | <0.001 |
| Concomitant pain | 3 | 637 | 1.83 (1.25–2.67) | | | 3.10 | <0.001 |
| Disturbed sleep since onset | 3 | 570 | 2.96 (0.97–9.04) | | | 1.90 | 0.06 |
| Cold hyperalgesia | 3 | 315 | 1.36 (0.91–2.05) | | | 1.50 | 0.133 |
| Higher age | 12 | 2347 | 1.00 (0.97–1.04) | | | 0.07 | 0.94 |
| Higher BMI | 3 | 559 | 1.24 (0.71–2.19) | | | 0.76 | 0.45 |
| Smoking | 4 | 38,188 | 1.24 (1.14–1.34) | | | 5.120 | <0.001 |
| **Meta-analysis (LR-/LR+)** | **Studies** | **Sample** | **LR- (95% CI)** | **Z** | **P** | **LR+ (95% CI)** | **Z** | **P** |
| Fear avoidance | 5 | 4621 | 0.50 (0.35–0.71) | -3.95 | <0.001 | 2.11 (1.59–2.80) | 5.20 | <0.001 |
| Poorer psychological health | 7 | 6200 | 0.77 (0.70–0.85) | -5.34 | <0.001 | 1.92 (1.67–2.18) | 9.50 | <0.001 |
| Somatisation | 3 | 2945 | 0.63 (0.48–0.82) | -3.37 | <0.001 | 2.56 (1.72–3.82) | 4.60 | <0.001 |
| Lower job satisfaction | 5 | 1888 | 0.95 (0.90–1.01) | -1.61 | 0.108 | 1.35 (1.05–1.74) | 2.32 | 0.020 |
| Higher job demands | 4 | 4059 | 0.86 (0.82–0.91) | -5.35 | <0.001 | 1.30 (1.12–1.51) | 3.44 | <0.001 |
| Lower socioeconomic status | 10 | 7008 | 0.78 (0.68–0.90) | -3.49 | <0.001 | 1.06 (1.02–1.10) | 2.74 | 0.006 |
| Financial compensation | 7 | 2786 | 0.87 (0.80–0.95) | -3.23 | 0.001 | 1.48 (1.24–1.76) | 4.38 | <0.001 |
| High levels of pain at or near onset | 8 | 6260 | 0.51 (0.38–0.68) | -4.55 | <0.001 | 1.69 (1.39–2.04) | 5.34 | <0.001 |
| Higher levels of functional impairment | 8 | 6888 | 0.40 (0.26–0.61) | -4.28 | <0.001 | 1.88 (1.40–2.51) | 4.22 | <0.001 |
| Female sex/gender | 16 | 8470 | 0.92 (0.85–0.99) | -2.19 | <0.029 | 1.14 (1.04–1.26) | 2.78 | 0.005 |
| Higher age | 10 | 4899 | 0.94 (0.89–1.00) | -0.21 | 0.036 | 1.12 (1.01–1.24) | 2.16 | 0.031 |
| Poorer general health | 7 | 5431 | 0.84 (0.77–0.91) | -4.22 | <0.001 | 1.51 (1.27–1.80) | 4.68 | <0.001 |
| High BMI | 3 | 2237 | 1.02 (0.90–1.16) | 0.31 | 0.754 | 0.91 (0.73–1.14) | -0.81 | 0.418 |
| Smoking | 6 | 3007 | 0.91 (0.86–0.96) | -3.40 | <0.001 | 1.18 (1.08–1.30) | 3.64 | <0.001 |
| History of the same MSK pain | 9 | 3902 | 0.84 (0.72–0.98) | -2.24 | 0.025 | 1.08 (1.02–1.15) | 2.45 | 0.014 |

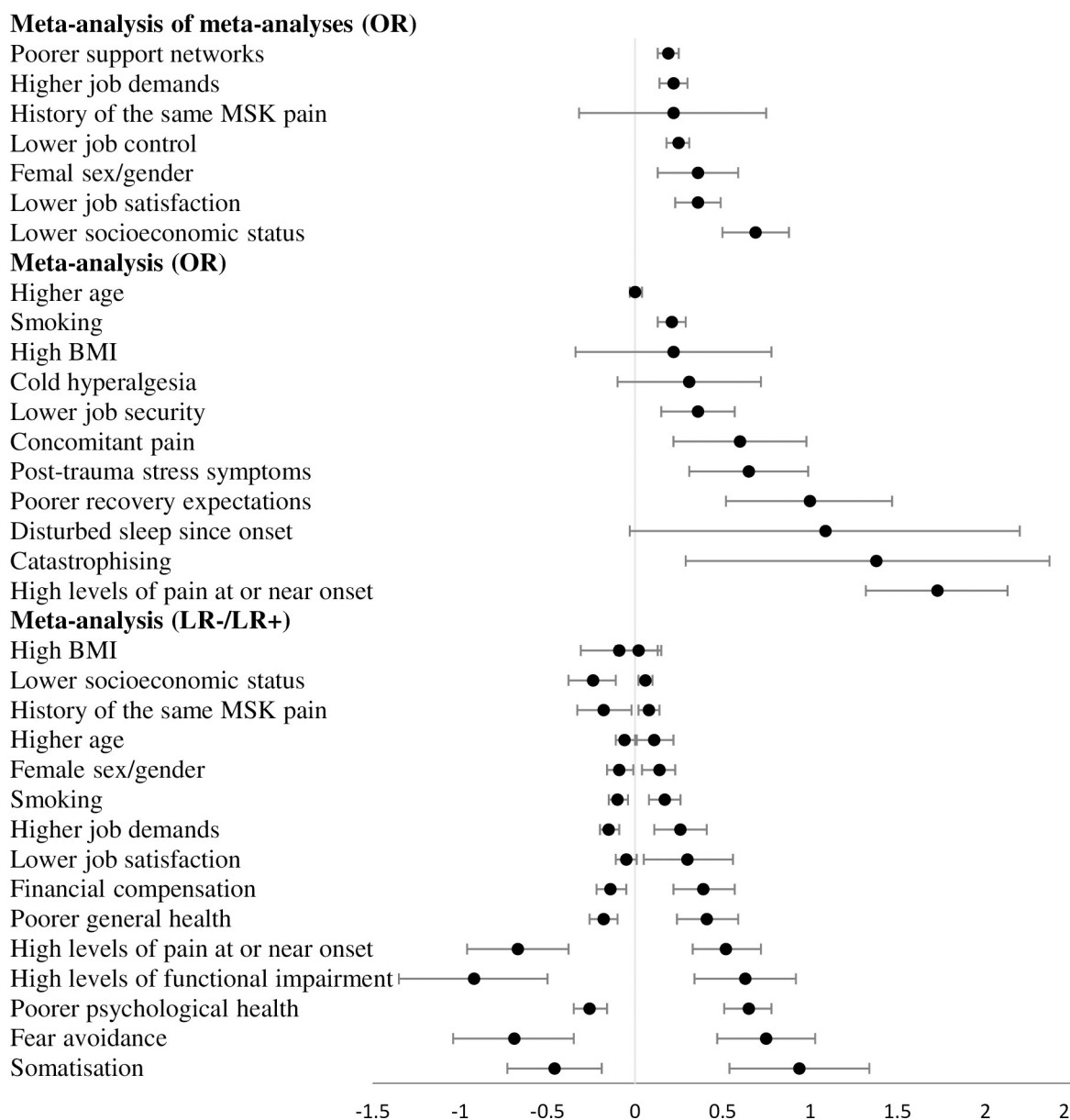

**Fig 2. Forest plot of meta-analyses on the natural log scale.** Presented on the natural log scale, where the number of null effect is 0 rather than 1, to ensure symmetrical representation of upper and lower 95% confidence intervals.

findings for each factor is presented in S4 Table. Eleven factors were supported by only one included review and therefore GRADE assessment and descriptive synthesis was not possible for these factors. See S5 Table for the primary data we used for all meta-analyses we performed.

## Risk of bias

Risk of bias assessment with AMSTAR-2 revealed that one review was low risk of bias [42] and 12 were high risk of bias [39–41, 43–51]. See S6 Table for full rating details. The main concerns were lack of *a priori* protocol registration/design [39, 41, 43–45, 47, 48, 50, 51], no justification

**Table 3. GRADE evidence profile and summary of findings.**

| № of reviews (№ of studies) | GRADE certainty assessment | | | | | Summary | | | Meta-analysis (95% CI) | GRADE level of Certainty |
|---|---|---|---|---|---|---|---|---|---|---|
| | Risk of bias | Inconsistency | Indirectness | Imprecision | Publication bias | № of participants | RoB of primary studies | Type of MSK pain/ condition | | |
| **Smoking** (physical health factors) | | | | | | | | | | |
| 2 (10) | Not serious | Not serious | Not serious | Not serious | Serious | 41,195 | Low | Lower back, neck | OR 1.24 (1.14–1.34); LR+ 1.18 (1.08–1.3); LR- 0.91 (0.86–0.96) | ⊕⊕⊕⊖ Moderate |
| **Fear avoidance** (psychological factors) | | | | | | | | | | |
| 3 (9) | Serious | Not serious | Not serious | Not serious | Not serious | 5208 | Low | Lower back, shoulder | LR+ 2.11 (1.59–2.8); LR- 0.5 (0.35–0.71) | ⊕⊕⊕⊖ Moderate |
| **Poorer support networks** (psychosocial factors) | | | | | | | | | | |
| 3 (71) | Serious | Not serious | Not serious | Not serious | Not serious | 95,738 | Low | Lower back, neck and/or shoulder, upper extremity, lower extremity, shoulder | OR* 1.21 (1.14–1.29) | ⊕⊕⊕⊖ Moderate |
| **Lower socioeconomic status** (psychosocial factors) | | | | | | | | | | |
| 4 (23) | Serious | Not serious | Not serious | Not serious | Not serious | 44,968 | Low | Neck, lower back, shoulder | OR* 2.0 (1.64–2.42); LR+ 1.06 (1.02–1.1); LR- 0.78 (0.68–0.90) | ⊕⊕⊕⊖ Moderate |
| **High levels of pain at or near onset** (symptoms or experiences at or near onset) | | | | | | | | | | |
| 3 (22) | Serious | Not serious | Not serious | Not serious | Not serious | 9394 | Low | Lower back, whiplash associated disorder | OR 5.61 (3.74–8.43); LR+ 1.69 (1.39–2.04); LR- 0.51 (0.38–0.68) | ⊕⊕⊕⊖ Moderate |
| **Poorer general health (physical health factors)** | | | | | | | | | | |
| **2 (9)** | Very serious | Not serious | Not serious | Not serious | Not serious | 6409 | Low | Lower back | LR+ 1.51 (1.27–1.8); LR- 0.84 (0.77–0.91) | ⊕⊕⊖⊖ Low |
| **Somatisation** (psychological factors) | | | | | | | | | | |
| 3 (8) | Serious | Not serious | Not serious | Not serious | Serious | 4742 | Low Moderate | Lower back, shoulder | LR+ 2.56 (1.72–3.82); LR- 0.63 (0.48–0.82) | ⊕⊕⊖⊖ Low |
| **Poorer psychological health** (psychological factors) | | | | | | | | | | |
| 3 (14) | Very serious | Not serious | Not serious | Not serious | Not serious | 9092 | Low | Lower back | LR+ 1.92 (1.67–2.18); LR- 0.77 (0.7–0.85) | ⊕⊕⊖⊖ Low |
| **Stress** (psychological factors) | | | | | | | | | | |
| 2 (2) | Serious | Not serious | Not serious | Serious | Not serious | 110 | High | Neck, lower back | N/A | ⊕⊕⊖⊖ Low |
| **Lower job satisfaction** (psychosocial factors) | | | | | | | | | | |
| 3 (48) | Very serious | Not serious | Not serious | Not serious | Not serious | 61,835 | Low | Lower back, neck and/or shoulder, upper extremity, lower extremity | OR* 1.43 (1.25–1.63); LR+ 1.35 (1.05–1.74); LR- 0.95 (0.9–1.01) | ⊕⊕⊖⊖ Low |
| **Financial compensation** (psychosocial factors) | | | | | | | | | | |
| 2 (11) | Very serious | Not serious | Not serious | Not serious | Not serious | 6085 | Low | Lower back | LR+ 1.48 (1.24–1.76); LR- 0.87 (0.8–0.95) | ⊕⊕⊖⊖ Low |
| **Concomitant pain** (symptoms or experiences at or near onset) | | | | | | | | | | |

*(Continued)*

**Table 3.** (Continued)

| № of reviews (№ of studies) | GRADE certainty assessment | | | | | Summary | | | Meta-analysis (95% CI) | GRADE level of Certainty |
|---|---|---|---|---|---|---|---|---|---|---|
| | Risk of bias | Inconsistency | Indirectness | Imprecision | Publication bias | № of participants | RoB of primary studies | Type of MSK pain/ condition | | |
| 2 (4) | Serious | Not serious | Not serious | Not serious | Serious | 547 | Moderate | Lower back, whiplash associated disorder | OR 1.83 (1.25–2.67) | ⊕⊕⊖⊖ Low |
| **Higher levels of functional impairment at onset** (symptoms or experiences at or near onset) | | | | | | | | | | |
| 2 (16) | Very serious | Not serious | Not serious | Not serious | Not serious | 11,654 | Low | Lower back | LR+ 1.88 (1.4–2.51); LR- 0.4 (0.26–0.61) | ⊕⊕⊖⊖ Low |
| **Time off work** (symptoms or experiences at or near onset) | | | | | | | | | | |
| 2 (7) | Serious | Not serious | Not serious | Serious | Not serious | 4681 | Moderate | Lower back, shoulder | N/A | ⊕⊕⊖⊖ Low |
| **History of the same MSK pain** (physical health factors) | | | | | | | | | | |
| 4 (30) | Serious | Serious | Not serious | Serious | Serious | 9292 | Moderate | Lower back, whiplash associated disorder, shoulder | OR* 1.24 (0.73–2.12); LR+ 1.08 (1.02–1.15); LR- 0.84 (0.72–0.98) | ⊕⊖⊖⊖ Very low |
| **High BMI** (not associated) (physical health factors) | | | | | | | | | | |
| 2 (7) | Serious | Not serious | Not serious | Serious | Not serious | 2796 | Low | Lower back, whiplash associated disorder | OR 1.24 (0.71–2.19) LR+ 0.91 (0.73–1.14); LR- 1.02 (0.9–1.16) | ⊕⊖⊖⊖ Very low |
| **Depression** (psychological factors) | | | | | | | | | | |
| 2 (5) | Serious | Not serious | Serious | Serious | Not serious | 917 | Moderate | Lower back, shoulder | N/A | ⊕⊖⊖⊖ Very low |
| **Catastrophising** (psychological factors) | | | | | | | | | | |
| 3 (7) | Serious | Not serious | Not serious | Serious | Serious | 1050 | Moderate | Lower back, shoulder, whiplash associated disorder | OR 3.99 (1.33–10.74) | ⊕⊖⊖⊖ Very low |
| **Poorer coping strategies** (psychological factors) | | | | | | | | | | |
| 2 (7) | Serious | Not serious | Serious | Serious | Serious | 1875 | Moderate | Lower back, shoulder | N/A | ⊕⊖⊖⊖ Very low |
| **Higher job demands** (psychosocial factors) | | | | | | | | | | |
| 4 (75) | Very serious | Not serious | Serious | Not serious | Not serious | 110, 609 | Low | Lower back, neck and/or shoulder, upper extremity, shoulder | OR* 1.25 (1.15–1.35); LR+ 1.3 (1.12–1.51); LR- 0.86 (0.82–0.91) | ⊕⊖⊖⊖ Very low |
| **Lower job control** (psychosocial factors) | | | | | | | | | | |
| 2 (48) | Very serious | Not serious | Serious | Not serious | Not serious | 74,200 | High | Lower back, neck and/or shoulder, upper extremity, lower extremity, shoulder | OR* 1.28 (1.2–1.37) | ⊕⊖⊖⊖ Very low |
| **Making physical compensations** (symptoms or experiences at or near onset) | | | | | | | | | | |
| 2 (5) | Serious | Serious | Serious | Serious | Not serious | 167 | High | Lower back, shoulder | N/A | ⊕⊖⊖⊖ Very low |
| **Female sex/gender** (demographic factors) | | | | | | | | | | |
| 4 (38) | Serious | Serious | Serious | Not serious | Not serious | 15,982 | Moderate | Lower back, whiplash associated disorder | OR* 1.43 (1.13–1.81); LR+ 1.14 (1.04–1.26); LR- 0.92 (0.85–0.99) | ⊕⊖⊖⊖ Very low |
| **Higher age** (demographic factors) | | | | | | | | | | |

(Continued)

**Table 3.** (Continued)

| № of reviews (№ of studies) | GRADE certainty assessment | | | | | Summary | | | Meta-analysis (95% CI) | GRADE level of Certainty |
|---|---|---|---|---|---|---|---|---|---|---|
| | Risk of bias | Inconsistency | Indirectness | Imprecision | Publication bias | № of participants | RoB of primary studies | Type of MSK pain/ condition | | |
| 4 (26) | Serious | Serious | Not serious | Serious | Not serious | 34,802 | Moderate | Lower back, shoulder, whiplash associated disorder | OR 1.0 (0.97–1.04); LR+ 1.12 (1.01–1.24); LR-0.94 (0.89–1.0) | ⊕⊖⊖⊖ Very low |
| **Post trauma stress symptoms** (psychological factors) | | | | | | | | | | |
| 1 (7) | N/A | N/A | N/A | N/A | N/A | 1695 | Low | Whiplash associated disorder | OR 1.92 (1.37–2.69) | N/A |
| **Stressful childhood experiences** (psychological factors) | | | | | | | | | | |
| 1 (1) | N/A | N/A | N/A | N/A | N/A | 9552 | High | Lower back | N/A | N/A |
| **Poorer recovery expectations** (psychological factors) | | | | | | | | | | |
| 1 (6) | N/A | N/A | N/A | N/A | N/A | 2514 | Low | Lower back | OR 2.72 (1.68–4.35) | N/A |
| **Lower job security** (psychosocial factors) | | | | | | | | | | |
| 1 (8) | N/A | N/A | N/A | N/A | N/A | 11,817 | High | Lower back | OR 1.43 (1.16–1.76) | N/A |
| **Higher domestic responsibilities** (psychosocial factors) | | | | | | | | | | |
| 1 (2) | N/A | N/A | N/A | N/A | N/A | Not stated | High | Lower back | N/A | N/A |
| **Dissatisfaction during leisure activities** (psychosocial factors) | | | | | | | | | | |
| 1 (1) | N/A | N/A | N/A | N/A | N/A | Not stated | High | Lower back | N/A | N/A |
| **Being divorced or widowed without children** (psychosocial factors) | | | | | | | | | | |
| 1 (1) | N/A | N/A | N/A | N/A | N/A | Not stated | High | Lower back | N/A | N/A |
| **Disturbed sleep since onset** (symptoms or experiences at or near onset) | | | | | | | | | | |
| 1 (3) | N/A | N/A | N/A | N/A | N/A | 570 | Moderate | Whiplash associated disorder | OR 2.96 (0.97–9.04) | N/A |
| **Cold hyperalgesia** (symptoms or experiences at or near onset) | | | | | | | | | | |
| 1 (6) | N/A | N/A | N/A | N/A | N/A | 443 | High | Whiplash associated disorder | OR 1.36 (0.91–2.05) | N/A |
| **Sudden onset** (symptoms or experiences at or near onset | | | | | | | | | | |
| 1 (1) | N/A | N/A | N/A | N/A | N/A | Not stated | High | Lower back | N/A | N/A |
| **Lack of energy** (symptoms or experiences at or near onset) | | | | | | | | | | |
| 1 (1) | N/A | N/A | N/A | N/A | N/A | Not stated | High | Lower back | N/A | N/A |

for excluding studies [39, 41, 43, 47, 48], inadequate/no assessment of risk of bias of primary studies [41, 43, 45, 48] or did not consider risk of bias in interpretation of findings [41, 43, 45, 47–49]. Risk of bias assessment of the summarised primary studies revealed 40% are low risk of bias, 26% are moderate, and 34% are high risk of bias.

## GRADE certainty of evidence

There is moderate certainty evidence that smoking, fear avoidance, poorer support networks, lower socioeconomic status, and high levels of pain at or near onset, are associated with development of CMP (all P<0.001). There is low certainty evidence that poorer general health, somatisation, poorer psychological health, lower job satisfaction, financial compensation, concomitant pain, and higher levels of functional impairment (all P<0.001); as well as stress and time off work (supported by descriptive synthesis only). Remaining factors are very low certainty. The main reason for downrating evidence was high risk of bias of reviews (see Table 3).

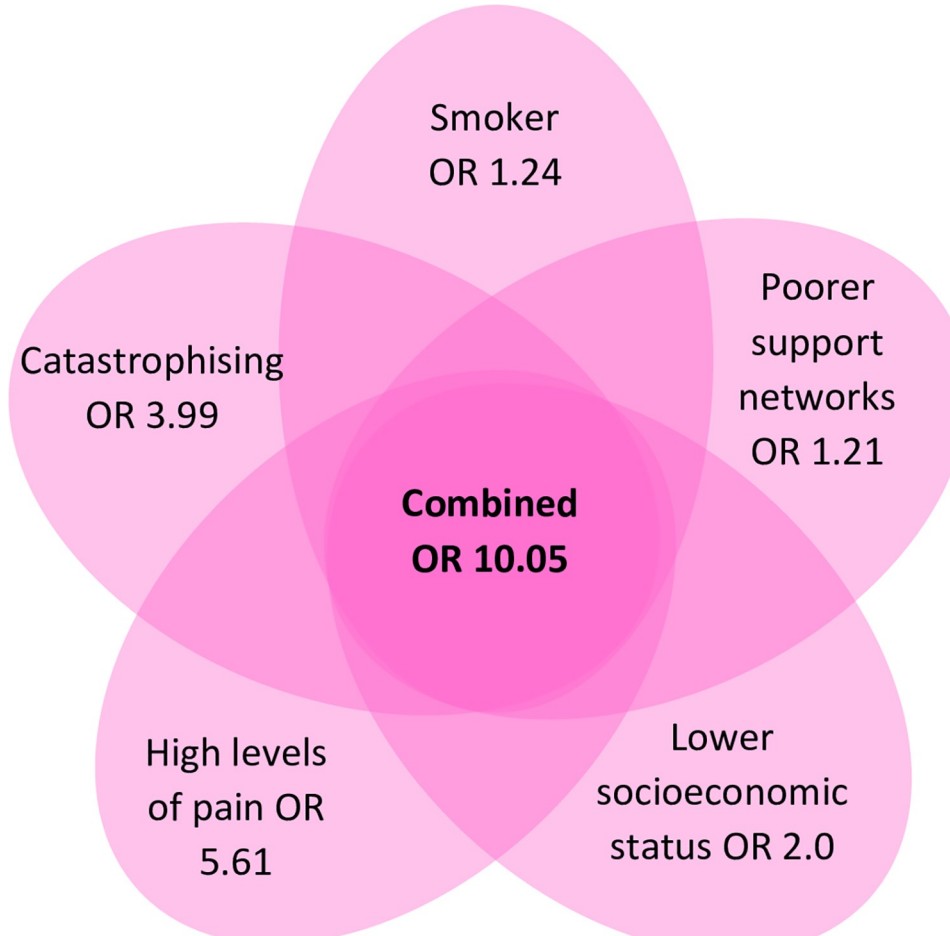

**Fig 3. Theoretical combined odds ratio for development of CMP.** Based on risk aggregation methods [69]. Formula: (OR) + (OR) + (OR) + (OR) + (OR)–(total number of odds ratios) + 1.0.

## Discussion

To the best of the authors knowledge, this umbrella review is the largest synthesis of research of biopsychosocial factors that contribute to development of CMP. This review also provides the first published guidance of how to apply GRADE for the purpose of an umbrella review. GRADE is widely considered as a best practice framework which provides a systematic approach to determine the quality of evidence and making clinical practice recommendations. Despite this, it has not been routinely adopted for use in umbrella reviews, likely because no clear methodological guidance exists. Our Methods for Application of GRADE for an Epidemiological Umbrella Review (S2 File) seeks to reconcile this discrepancy between systematic and umbrella reviews and may serve as guidance for future umbrella reviews.

The key findings of this umbrella review are that there is moderate level evidence that smoking, fear avoidance, poorer support networks, lower socioeconomic status, and high levels of pain at or near onset are associated with development of CMP. These findings are pertinent to a great number of stakeholders worldwide including healthcare policymakers, clinical decision makers, researchers, patients and the public. CMP is the leading cause of disability globally [52] with substantial impact on quality of life for individuals, and loss of productivity/ burden on healthcare services and society [6, 9]. The 34 biopsychosocial factors identified

within this review that contribute to development of CMP are not related to a specific MSK condition, but rather are characteristics of the person or their experience with pain. These characteristics and experiences often exist independently of any structural 'abnormality' that may have been diagnosed and targeted as part of condition-centred management approaches. This suggests that traditional understandings of mechanisms of MSK pain, its management, and its chronicity, are likely an oversimplification of an evidently complex phenomenon. This may explain why MSK condition-centred approaches conventionally utilised by healthcare services are proving inadequate, with the prevalence and burden of CMP rising [2, 11].

CMP may be explained by alterations of the nociceptive pain systems leading to continued or repeated experience of pain even with little or no evidence of potential or actual tissue damage; this is termed 'nociplastic pain' [53]. This arises due to functional and anatomical alterations within the central nervous system whereby there a shift of activity from the somatosensory cortex to the corticolimbic system [54]. This area of the brain is important for emotional contextualisation, reward anticipation, stress response, decision making, memory modulation, and movement behaviours [55–57]. These functions are utilised to scrutinise nociceptive and sensory input and establishing protective behaviours such as fear, stress, and avoidance in response [57–60]. Further to this, the body's natural pain-relieving mechanisms such as descending inhibition are diminished [61] with increased neurotransmission of nociceptive action potentials at the dorsal horn (central sensitisation) [62] and increased production of sensitising chemical mediators at both the dorsal horn and peripheral nociceptive nerve endings (peripheral sensitisation) [62, 63], thus facilitating a nervous system which is wholly sensitised and geared towards efficiently and frequently producing the experience of pain–CMP. This may explain how sham interventions work to improve CMP if, for example, this creates a positive emotional experience for patients such as hopefulness and reassurance within the corticolimbic system [64], reducing stress responses and increasing reward anticipation [65, 66]; thus re-activating descending inhibition [67] and shifting away from the increased perception of threat. Recognition of these nociplastic mechanisms and the biopsychosocial factors that perpetuate them (as identified within this review) presents an opportunity for healthcare services to better manage people with MSK pain. However, the efficacy of such approaches are likely to be highly influenced by patients beliefs about the cause of their MSK pain which, given traditional healthcare approaches, are likely to be condition-centred. A widescale shift toward patient-centred management and away from condition-centred approaches may therefore be beneficial to better managing CMP.

Meta-analysis was possible for 25 biopsychosocial factors with effect sizes/magnitude of effect possibly perceived as small for most biopsychosocial factors [68]. However, it is unlikely that one overarching factor leads to development of CMP for affected individuals, but rather the combination of multiple factors. Risk aggregation methods, whereby the overall risk is considered the sum of individual risks [69], can be utilised to demonstrate a theoretical example of the combined odds of developing CMP in the presence of multiple biopsychosocial factors (see Fig 3). In this example, five common biopsychosocial factors are combined creating an aggregated odds ratio of 10.05 for development of CMP, which is considered a very large increase in risk [68]. This combination of factors is reflective of many individuals who may present to healthcare settings with MSK pain who, based on our combined odds ratio, may be over 10 times more likely to develop CMP than an individual who does not share this presentation. The presence of many of these factors will be influenced by the unique backgrounds, experiences and beliefs of individuals. This further demonstrates the need for person-centred assessment and management approaches.

## Strengths and limitations

The main strengths are that the protocol for this research was designed, peer reviewed, and published *a priori* [31], the methods used are underpinned by validated frameworks such as Cochrane/Joanna Briggs guidance and the PRISMA checklist for design and reporting research, certainty of evidence was established through GRADE; and this umbrella review fulfils 'high confidence in findings' criteria of AMSTAR-2. There are also limitations which require consideration. Many of the identified biopsychosocial factors are of low to very low certainty evidence, mostly due to risk of bias of reviews. However, this is not a reflection of the quality of included primary studies of which 66% were low to moderate risk of bias. Furthermore, 141 potentially eligible reviews retrieved for full text screening were excluded due to the inclusion of methods such as cross sectional or case control designs. These methods are ill equipped to distinguish between cause and effect [70] and therefore are not appropriate for determining factors that *contribute to* development of CMP, rather than *caused by* CMP.

## Recommendations for further research

This umbrella review highlights high risk of bias within existing systematic reviews which seek to identify factors that contribute to development of CMP and therefore further systematic reviews are required to improve upon the certainty of findings presented. Future systematic reviews should be informed by validated published guidance and should use observational studies of prospective longitudinal cohorts only to ensure synthesis of reliable findings.

## Conclusion

Findings identified 34 biopsychosocial factors associated with development of CMP, and one factor that was not associated. These findings are situated within five domains: physical health, psychological factors, psychosocial factors, symptoms or experiences at or near onset, and demographics. Smoking, fear avoidance, poorer support networks, lower socioeconomic status, and high levels of pain at or near onset are supported by moderate certainty evidence. Although the remaining factors identified are of low to very low certainty evidence, many of these findings are compelling due to the consistency of findings across included reviews and low to moderate risk of bias of primary studies for most factors. The factors associated with development of CMP are in keeping with nociplastic mechanisms of pain and support the need for a paradigm shift in healthcare management of CMP with less focus on MSK structures and more focus on broader biopsychosocial health. As such, it would be sensible for policymakers and clinical decision makers to incorporate these findings into clinical practice by adopting a more person-centred than condition-centred approach to assessing and treating people with MSK pain. However, further high-quality systematic reviews are recommended to increase certainty of evidence of these findings.

## Supporting information

**S1 File. PRISMA checklist.**
(DOCX)

**S2 File. GRADE guidance.**
(DOCX)

**S1 Table. Search strategy.**
(DOCX)

**S2 Table. Review characteristics.**
(DOCX)

**S3 Table. Definitions of domains and factors.**
(DOCX)

**S4 Table. Descriptive synthesis.**
(DOCX)

**S5 Table. Primary data used in meta-analysis.**
(DOCX)

**S6 Table. AMSTAR-2 risk of bias assessment.**
(DOCX)

## Acknowledgments

Stephen Reid, Liaison Librarian at St. George's University of London provided expert feedback and guidance on the design of our literature search strategy. Dr Farid Foroutan is the lead author of GRADE guidance for prognostic factors in systematic reviews and provided invaluable feedback for our adaptation of GRADE guidelines for this umbrella review.

## Author Contributions

**Conceptualization:** Michael Dunn, Alison B. Rushton, Andrew Soundy, Nicola R. Heneghan.

**Data curation:** Michael Dunn.

**Formal analysis:** Michael Dunn, Alison B. Rushton, Jai Mistry, Andrew Soundy, Nicola R. Heneghan.

**Investigation:** Michael Dunn, Jai Mistry.

**Methodology:** Michael Dunn, Alison B. Rushton, Jai Mistry, Andrew Soundy, Nicola R. Heneghan.

**Supervision:** Alison B. Rushton, Andrew Soundy, Nicola R. Heneghan.

**Visualization:** Michael Dunn, Alison B. Rushton, Andrew Soundy, Nicola R. Heneghan.

**Writing – original draft:** Michael Dunn.

**Writing – review & editing:** Michael Dunn, Alison B. Rushton, Jai Mistry, Andrew Soundy, Nicola R. Heneghan.

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
