## [Decision Letter · Decision Letter 0]

12 Feb 2024

PONE-D-23-34953The biopsychosocial factors associated with development of chronic musculoskeletal pain. An umbrella review and meta-analysis of observational systematic reviewsPLOS ONE

Dear Dr. Dunn,

Thank you for submitting your manuscript to PLOS ONE. After careful consideration, we feel that it has merit but does not fully meet PLOS ONE’s publication criteria as it currently stands. Therefore, we invite you to submit a revised version of the manuscript that addresses the points raised during the review process.

**ACADEMIC EDITOR: Dear Author, Please attend to all the reviewers comments and make the necessary corrections.** The decision of this manuscript is justified based on PLOS ONE’s publication criteria and not on its novelty or perceived impact.

We look forward to receiving your revised manuscript.

Kind regards,

Zulkarnain Jaafar

Academic Editor

PLOS ONE

Reviewers' comments:

Reviewer's Responses to Questions

**Comments to the Author**

1. Is the manuscript technically sound, and do the data support the conclusions?

Reviewer #1: Yes

Reviewer #2: Partly

2. Has the statistical analysis been performed appropriately and rigorously? 

Reviewer #1: Yes

Reviewer #2: Yes

3. Have the authors made all data underlying the findings in their manuscript fully available?

Reviewer #1: Yes

Reviewer #2: Yes

4. Is the manuscript presented in an intelligible fashion and written in standard English?

Reviewer #1: Yes

Reviewer #2: Yes

5. Review Comments to the Author

Reviewer #1: Please make the eligibility criteria clearer (study design, duration, location,etc) needs to be added

Take care of grammars and check the spelling of certain words which needs correction

References should be in proper format

Reviewer #2: The paper titled "The biopsychosocial factors associated with the development of chronic musculoskeletal pain: An umbrella review and meta-analysis of observational systematic reviews" provides a comprehensive analysis of biopsychosocial factors contributing to chronic musculoskeletal pain. The paper aims to establish which factors are associated with this condition by reviewing and meta-analyzing existing systematic reviews. The research utilized a wide range of databases to collect data up until April 4th, 2023, and included 13 systematic reviews comprising 185 original studies with 489,644 participants. It found moderate certainty evidence that smoking, fear avoidance, poorer support networks, lower socioeconomic status, and high levels of pain are associated with the development of chronic musculoskeletal pain. However, the study also highlights a high risk of bias in most included reviews, suggesting a need for higher quality systematic reviews in the future.

Your manuscript does not meet the criteria for a systematic review, it should present clearly answerable questions and answering those questions should mean that the questions are controversial and can only be resolved by considering a large number of papers with substantive data. The idea that biopsychosocial factors contribute to chronic musculoskeletal pain is not controversial – this topic is better suited as a “chapter review” in a book. Furthermore, the authors submitted this manuscript as a ‘research article’. Please change into ‘review article’.

Abstract:

• Aim: Please rephrase the aim into a full sentence.

Introduction

Lines 54-73:

• This section effectively sets the stage for the importance of understanding biopsychosocial factors in CMP. However, consider adding a sentence that directly states the gap in the current literature that this study aims to fill, tying directly to the purpose of conducting an umbrella review.

Lines 74-90:

• The transition from acute to chronic MSK pain and the role of biopsychosocial factors is well articulated. To strengthen this section, a brief mention of the theoretical framework guiding the investigation of these factors could provide more depth.

Methods:

Aim: Please rephrase the aim into a full sentence.

Lines 112-127:

• The exclusion criteria are comprehensive but consider specifying if there were any limitations based on study quality, which can impact the scope of the review.

Lines 128-137:

• Clarify if language restrictions were applied and how grey literature was incorporated into the analysis.

Lines 149-155:

• The use of AMSTAR 2 for risk of bias assessment is appropriate. Clarify how discrepancies between reviewers were resolved to ensure transparency in the review process.

Lines 158-175:

• The methodological detail is good, but specifying the criteria for considering studies for meta-analysis versus descriptive synthesis could enhance clarity. Additionally, explain any methods used to assess heterogeneity among studies.

Lines 176-189:

• The adaptation of GRADE for umbrella reviews is innovative.

Results:

Lines 234-240:

• Clearly delineate the implications of the high risk of bias in included reviews. For example, discuss how this affects the reliability of the conclusions drawn and what steps future reviews could take to mitigate such biases.

Discussion:

Lines 250-256:

• Briefly summarize the key findings before delving into their implications. Highlight the novelty and significance of applying GRADE in this context more explicitly.

Conclusion:

Lines 326-338:

• Emphasize the need for a paradigm shift in MSK pain management more strongly. Outline steps for how healthcare providers can adopt a person-centered approach based on the review's findings.

6. PLOS authors have the option to publish the peer review history of their article (what does this mean?). If published, this will include your full peer review and any attached files.

Reviewer #1: No

Reviewer #2: No

---

## [Author Response · Author response to Decision Letter 0]

11 Mar 2024

1) Journal requirements

Comments

Please ensure that your manuscript meets PLOS ONE's style requirements, including those for file naming. The PLOS ONE style templates can be found at https://journals.plos.org/plosone/s/file?id=wjVg/PLOSOne_formatting_sample_main_body.pdf and https://journals.plos.org/plosone/s/file?id=ba62/PLOSOne_formatting_sample_title_authors_affiliations.pdf.

Authors’ response

We have checked and ensured the manuscript meets the style requirements including for file naming.

2) Reviewer 1 comments

Reviewer 1 comment

Please make the eligibility criteria clearer (study design, duration, location,etc) needs to be added Thank you for your feedback we have considered your comments in full. 

Authors’ response

The inclusion criteria is outlined via the PICOS framework as per Cochrane guidance for the inclusion criteria of a systematic review (Thomas et al, 2023). Study design was already outlined as: “systematic reviews, with or without meta-analysis, of prospective observational longitudinal studies” under ‘S’ of the PICOS for study design. We have additionally added “Original studies included in reviews must have been at least three months in duration with no limitations on the study setting”. Thank you for this observation, we hope this is clearer. Please let us know if you have any further specific information you think should be included. 

Reference

Thomas J, Kneale D, McKenzie JE, Brennan SE, Bhaumik S. Chapter 2: Determining the scope of the review and the questions it will address. In: Higgins JPT, Thomas J, Chandler J, Cumpston M, Li T, Page MJ, Welch VA (editors). Cochrane Handbook for Systematic Reviews of Interventions version 6.4 (updated August 2023). Cochrane, 2023. Available from www.training.cochrane.org/handbook.

Reviewer 1 comment

Take care of grammars and check the spelling of certain words which needs correction

Authors’ response

Thank you for your feedback. We have re-reviewed the manuscript to check for grammatical and spelling errors. Please note that this manuscript is written in ‘British English’ language and therefore some spellings may differ for those more familiar with ‘American English’ e.g., catastophisation vs catastrophization. 

Reviewer 1 comment

References should be in proper format

Authors’ response

References have been automatically formatted in “Vancouver” style, in keeping with Plos One submission guidance using Endnote software. We have also manually screened the referencing throughout the manuscript and the reference list and corrected where applicable. We are not aware of any errors currently with the reference formatting. Please do make us aware of any errors and we will amend accordingly.

3) Reviewer 2 comments

Reviewer 2 comment

Your manuscript does not meet the criteria for a systematic review, it should present clearly answerable questions and answering those questions should mean that the questions are controversial and can only be resolved by considering a large number of papers with substantive data. The idea that biopsychosocial factors contribute to chronic musculoskeletal pain is not controversial – this topic is better suited as a “chapter review” in a book. 

Authors’ response

Thank you very much for these observations. We base our rationale for this systematic review on these underlying principles – we have summarised a large amount of information and identified key findings useful for clinicians, researchers, and policymakers. These underlying principles are endorsed by Cochrane guidance for “Starting a Review” who describe that the primary purpose of a systematic review is “to inform people making decisions about health or health care” (Lasserson et al, 2023). We therefore feel the submission of this work as a systematic review is appropriate as these findings are likely to influence decision making. Namely, clinical decision and policymaking with less weighting to specific MSK structural management, and more weighting to holistic determinants or health incorporating the biopsychosocial factors outlined in our review. 

Furthermore, Cochrane also state other key points to meet the high standards expected of a systematic review to include:

- Minimisation of risk of bias with pre-specified research questions and methods that are documented in protocols

- Conducted by a team that includes domain expertise and methodological expertise

- Good data management and quality assurance mechanisms

We believe we have also fulfilled these requirements. Our protocol was designed, peer reviewed and published a priori. Our authorship team experience covers all key domains of this umbrella review including over 60 years combined clinical MSK experience and several published systematic and umbrella reviews with and without meta-analyses.

Based on this published Cochrane guidance, we feel we have fully met the requirements for publication of this research as a systematic review of systematic reviews, also known as an umbrella review or overview of reviews. Please do let us know if you have any further concerns about this. 

References

Lasserson TJ, Thomas J, Higgins JPT. Chapter 1: Starting a review. In: Higgins JPT, Thomas J, Chandler J, Cumpston M, Li T, Page MJ, Welch VA (editors). Cochrane Handbook for Systematic Reviews of Interventions version 6.4 (updated August 2023). Cochrane, 2023. Available from www.training.cochrane.org/handbook.

Reviewer 2 comment

Furthermore, the authors submitted this manuscript as a ‘research article’. Please change into ‘review article’. 

Authors’ response

The Plos One submission guidelines state:

“If your article is a systematic review or a meta-analysis you should:

Select “Research Article” as your article type when submitting”

Available here: https://journals.plos.org/plosone/s/submission-guidelines under ‘Guidelines for specific study types’ and ‘Systematic reviews and meta-analyses’

Thanks for the suggestion, it’s always worth double checking these things. 

Reviewer 2 comment

Abstract:

• Aim: Please rephrase the aim into a full sentence. 

Authors’ response

This has been re-written as a full sentence. Thank you.

Reviewer 2 comment

Introduction

Lines 54-73:

This section effectively sets the stage for the importance of understanding biopsychosocial factors in CMP. However, consider adding a sentence that directly states the gap in the current literature that this study aims to fill, tying directly to the purpose of conducting an umbrella review. 

Authors’ response

We have added a sentence directly identifying the gap in the current literature and tied this to the study aims and the purpose of doing an umbrella review. This is located at lines 100-103 of the marked revised manuscript. We have kept this toward the end of the introduction for flow purposes. Thank you for this suggestion. 

Reviewer 2 comment

Lines 74-90:

• The transition from acute to chronic MSK pain and the role of biopsychosocial factors is well articulated. To strengthen this section, a brief mention of the theoretical framework guiding the investigation of these factors could provide more depth. 

Authors’ response

We aren’t fully clear of the meaning of this suggestion. Do you mean the research philosophy and theoretical framework underpinning observational methods? This would be better placed in the methods, but isn’t usually detailed within systematic review methodology. Please clarify and we will aim to resolve.

Reviewer 2 comment

Methods:

Aim: Please rephrase the aim into a full sentence. 

Authors’ response

This has been changed. Thank you. 

Reviewer 2 comment

Lines 112-127:

• The exclusion criteria are comprehensive but consider specifying if there were any limitations based on study quality, which can impact the scope of the review. 

Authors’ response

No limitations were placed on eligibility based on study quality. We have added a statement outlining this in the exclusion criteria section. Lines 133-134 of the marked manuscript. Thank you. 

Reviewer 2 comment

Lines 128-137:

• Clarify if language restrictions were applied and how grey literature was incorporated into the analysis. 

Authors’ response

Language restrictions were applied to include reviews available in the English language, or easily convertible to the English language using readily available software (e.g., google translate). This is outlined at lines 132-133 of the marked manuscript. 

Regarding the grey literature, no additional methods were used for incorporating any systematic reviews sourced through grey literature into the analysis. The methods detailed for eligibility and analysis were universally applied regardless of the source.

Thank you very much for these comments.

Reviewer 2 comment

Lines 149-155:

• The use of AMSTAR 2 for risk of bias assessment is appropriate. Clarify how discrepancies between reviewers were resolved to ensure transparency in the review process. 

Authors’ response

Discrepancies were referred to a third reviewer (NH). This is outlined at lines 159-160 of the marked manuscript. 

Reviewer 2 comment

Lines 158-175:

• The methodological detail is good, but specifying the criteria for considering studies for meta-analysis versus descriptive synthesis could enhance clarity. Additionally, explain any methods used to assess heterogeneity among studies. 

Authors’ response

We performed meta-analysis for any factor where there were at least two synthesisable sets of quantitative data e.g., two odds ratios for ‘stress’. This is outlined at lines 166-167 of the marked manuscript. Descriptive synthesis was performed for all factors regardless of meta-analysis having been performed or not. This is outlined at lines 181-182 of the marked manuscript. 

The methods used to assess heterogeneity are outlined in our Methods for the Application of GRADE (File S3 in the supplement) under the heading ‘Inconsistency’. Essentially, so long as findings were consistent across included reviews and the primary studies within reviews, then no inconsistency (heterogeneity) was determined to be present. Please see the ‘Inconsistency’ section in our GRADE methods (File S3) for full details.

Thank you very much for these comments.

Reviewer 2 comment

Lines 176-189:

• The adaptation of GRADE for umbrella reviews is innovative.

Authors’ response

Thank you! A lot of work went into that.

Reviewer 2 comment

Results:

Lines 234-240:

• Clearly delineate the implications of the high risk of bias in included reviews. For example, discuss how this affects the reliability of the conclusions drawn and what steps future reviews could take to mitigate such biases. 

Authors’ response

We agree this is very important. We have outlined these considerations within the ‘Strengths and Weaknesses’ and ‘Recommendations for Further Research’ sections of the discussion. We have stated that the high risk of bias of included reviews is the key reason the majority of factors identified within our review are low or very low certainty evidence outlined at lines 329-330 of the marked manuscript. We have also outlined that future systematic reviews should include only prospective longitudinal research and be informed by validated published guidance to ensure methodological rigour at lines 339-341 of the marked manuscript. Thank you. 

Reviewer 2 comment

Discussion:

Lines 250-256:

• Briefly summarize the key findings before delving into their implications. 

Authors’ response

We have described the key findings as per your suggestion at lines 262-264 of the marked manuscript. Thank you.

Reviewer 2 comment

Highlight the novelty and significance of applying GRADE in this context more explicitly. 

Authors’ response

We have made this more explicit, outlined at lines 261-267 of the marked manuscript. Thank you for this recommendation.

Reviewer 2 comment

Conclusion:

Lines 326-338:

• Emphasize the need for a paradigm shift in MSK pain management more strongly. Outline steps for how healthcare providers can adopt a person-centered approach based on the review's findings. 

Authors’ response

We have done this. Outlined at lines 350-358 of the marked manuscript. Thank you.

---

## [Decision Letter · Decision Letter 1]

18 Mar 2024

The biopsychosocial factors associated with development of chronic musculoskeletal pain. An umbrella review and meta-analysis of observational systematic reviews

PONE-D-23-34953R1

Dear Dr. Dunn,

We’re pleased to inform you that your manuscript has been judged scientifically suitable for publication and will be formally accepted for publication once it meets all outstanding technical requirements.

Kind regards,

Zulkarnain Jaafar

Academic Editor

PLOS ONE

Additional Editor Comments (optional):

Reviewers' comments:

Reviewer's Responses to Questions

**Comments to the Author**

1. If the authors have adequately addressed your comments raised in a previous round of review and you feel that this manuscript is now acceptable for publication, you may indicate that here to bypass the “Comments to the Author” section, enter your conflict of interest statement in the “Confidential to Editor” section, and submit your "Accept" recommendation.

Reviewer #1: All comments have been addressed

Reviewer #2: All comments have been addressed

2. Is the manuscript technically sound, and do the data support the conclusions?

Reviewer #1: Yes

Reviewer #2: Yes

3. Has the statistical analysis been performed appropriately and rigorously? 

Reviewer #1: Yes

Reviewer #2: Yes

4. Have the authors made all data underlying the findings in their manuscript fully available?

Reviewer #1: Yes

Reviewer #2: Yes

5. Is the manuscript presented in an intelligible fashion and written in standard English?

Reviewer #1: Yes

Reviewer #2: Yes

6. Review Comments to the Author

Reviewer #1: The article looks great and doesn't require any further changes. The required comments have been addressed and changes have been made.

Reviewer #2: The authors have addressed all the comments raised, providing thorough clarification on the discussed matters. Their umbrella review is conducted in a concise manner, summarizing the prevailing factors contributing to musculoskeletal pain. Furthermore, it offers valuable insights and implications for clinical application, making it a significant contribution to the field.

7. PLOS authors have the option to publish the peer review history of their article (what does this mean?). If published, this will include your full peer review and any attached files.

Reviewer #1: **Yes: **Naushaba Akhtar

Reviewer #2: No

---

## [Editor Report · Acceptance letter]

22 Mar 2024

PONE-D-23-34953R1 

PLOS ONE

Dear Dr. Dunn, 

I'm pleased to inform you that your manuscript has been deemed suitable for publication in PLOS ONE. Congratulations! Your manuscript is now being handed over to our production team.

Kind regards, 

on behalf of

Dr. Zulkarnain Jaafar 

Academic Editor

PLOS ONE